# DEF Cell-Derived Exosomal miR-148a-5p Promotes DTMUV Replication by Negative Regulating TLR3 Expression

**DOI:** 10.3390/v12010094

**Published:** 2020-01-14

**Authors:** Hongyan Guo, Anchun Cheng, Xingcui Zhang, YuHong Pan, Mingshu Wang, Juan Huang, Dekang Zhu, Shun Chen, Mafeng Liu, Xinxin Zhao, Ying Wu, Qiao Yang, Shaqiu Zhang, Yanling Yu, Leichang Pan, Bin Tian, Mujeeb Ur Rehman, Xiaoyue Chen, Yunya Liu, Ling Zhang, Zhongqiong Yin, Bo Jing, Renyong Jia

**Affiliations:** 1Research Center of Avian Disease, College of Veterinary Medicine, Sichuan Agricultural University, Chengdu 611130, China; 546406521@163.com (H.G.); 842975178@163.com (X.Z.); 474784959@163.com (Y.P.); mshwang@163.com (M.W.); huangjuan610@163.com (J.H.); zdk24@163.com (D.Z.); sophia_cs@163.com (S.C.); liumafengra@163.com (M.L.); xxinzhao@163.com (X.Z.); yingzi_no1@126.com (Y.W.); yangqiao721521@sina.com (Q.Y.); shaqiu86@hotmail.com (S.Z.); yanling3525@163.com (Y.Y.); pl2007@126.com (L.P.); btian_1985@163.com (B.T.); mujeebnasar@yahoo.com (M.U.R.); chenxy_24@sina.cn (X.C.); yunnyaaliu@163.com (Y.L.); zl97451@126.com (L.Z.); 2Institute of Preventive Veterinary Medicine, Sichuan Agricultural University, Chengdu 611130, China; 3Key Laboratory of Animal Disease and Human Health of Sichuan Province, Chengdu 611130, China; yinzhongq@163.com (Z.Y.); 305934258@163.com (B.J.)

**Keywords:** innate immunity, TLRs, DTMUV, exosome, miRNA

## Abstract

Duck tembusu virus (DTMUV) is a single-stranded, positive-polarity RNA flavivirus that has caused considerable economic losses in China in recent years. Innate immunity represents the first line of defense against invading pathogens and serves as an important role in resisting viral infections. In this study, we found that the infection of ducks by DTMUV triggers Toll-like receptors (TLRs) and (RIG-I)-like receptors (RLRs) signaling pathways and inducing abundant of pro-inflammatory factors and type I interferons (IFNs), in which melanoma differentiation-associated gene 5 (MDA5) and Toll-like receptor 3 (TLR3) play important immunity roles, they can inhibit the replication process of DTMUV via inducing type I IFNs. Moreover, we demonstrated that type I IFNs can inhibit the DTMUV replication process in a time- and dose-dependent manner. Exosomes are small membrane vesicles that have important roles in intercellular communication. MicroRNAs (miRNAs) are small non-coding RNAs that can modulate gene expression and are common substances in exosomes. In our experiment, we successfully isolated DEF cells derived exosome for the first time and explored its function. Firstly, we found the expression of miR-148a-5p is significantly decreased following DTMUV infect. Then we found miR-148a-5p can target TLR3 and down-regulate the expression of TLR3, serving as a negative factor in innate immunity. Unfortunately, we cannot find miRNAs with different expression changes that can target MDA5. Lastly, our experimental results showed that TLR3 was one of the causes of miR-148a-5p reduction, suggesting that the high level of TLR3 after DTMUV infect can both trigger innate immunity and suppress miR-148a-5p to resist DTMUV.

## 1. Introduction

Avian tembusu virus (ATMUV) is a single-stranded, positive-polarity RNA virus originally isolated from duck eggs. The ATMUV gene sequence has an open reading frame (ORF), three structural proteins (C, PrM, E) and seven non-structural proteins (NSl, NS2a, NS2b, NS3, NS4a, NS4b, and NS5) [1]. According to a previous study, a series of farms in southeast China first reported duck tembusu virus (DTMUV), after which it quickly spread northward from Beijing to Guangxi, ultimately resulting in huge economic losses [1]. Besides decreased egg production, DTMUV-infected animals manifest other clinical symptoms including anorexia, diarrhea, ataxia, and paralysis [2,3,4]. Additionally, severe ovarian hemorrhage, ovaritis, enlarged spleen and liver, necrosis of the spleen and necrosis in the brain are observed in DTMUV infected duck [1]. Morbidity in adult animals can range from 10%–30%, while morbidity in ducklings can reach 90%–100%, with a fatality rate of 5%–30% [2,3]. Although DTMUV was first identified in ducks, it has also been found in many other species such as geese [5], chicken [6], sparrow [7], and pigeons [8]. However, there are no existing reports of humans infected with the tembusu virus.

Innate immunity is the first line of defense against micro-organism invasion, playing essential roles in recognizing and clearing pathogens. Pattern recognition receptors (PRRs) are important immune receptors that identify conserved features of viruses and trigger innate immunity [9]. Toll-like receptors (TLRs) and retinoic acid-inducible gene I (RIG-I)-like receptors (RLRs) are two crucial types of PRRs. According to previous reports, they have important functions in resisting invasion by various viruses [9,10] including dengue [11], Japanese encephalitis, and Zika [12,13]. TLRs are type I transmembrane proteins that can bind leucine-rich repeats (LRRs) to recognize pathogen-associated molecular patterns (PAMPs) [14]. Then the intracellular Toll-interleukin 1 (IL-1) receptor (TIR) domain of TLRs interacts with a family of adaptor proteins to activate inflammatory responses and release inflammatory cytokines [14,15]. The TLR pathway consists of two branches based on whether the pathway contains the adaptation factor myeloid differentiation factor 88 (MyD88). TLR3 belongs to the TRIF pathway and can target the inflammatory response without MyD88 [14]. In the TRIF pathway, TLR3 activates TRIF to phosphorylate interferon (IFN) regulatory factor 3/7 (IRF3/7), which ultimately induces a number of inflammatory factors and type I IFN [16,17]. Activated TRIF also interacts with TRAF6 to target nuclear factor (NF)-κB to release inflammatory factors and induce chemotactic effects [18]. Besides TLR3, other TLRs target the MyD88 pathway to resist pathogen infection. In the MyD88 pathway, TLRs first interact with PAMPs. Following MyD88 recruitment, the IL-1 receptor associated kinases IRAK1, IRAK2, IRAK4, and IRAK-M trigger NF-κB and mitogen-activated protein kinases [19]. RIG-I and melanoma differentiation-associated gene 5 (MDA5) are two primary RLRs that can also induce type I IFN. Both RIG-I and MDA5 can recognize single- or double-stranded RNA viruses and then target mitochondrial antiviral-signaling protein (MAVS). MAVS, in turn, recruits TRAF3 to phosphorylate IRF3 and IRF7 and initiate the type I IFN antiviral immune response [20].

Chen et al. described the innate immune process that DTMUV induces in chicken [21], while Yu et al. detailed changes in the expression of inflammatory factors upon infection of duck with DTMUV [22]. Moreover, He Y et al. reveals the innate immune response to DTMUV infection in geese [5]. However, no study has addressed the specific mechanism underlying innate immunity against DTMUV in ducks. Thus we explored the specific immune processes triggered by DTMUV in ducks. Firstly, we examined important PRR family members and crucial inflammatory factors in ducklings and DEF cells following DTMUV infection. Significant up-regulation of PRRs, type I IFN, and some inflammatory factors were observed. Furthermore, MDA5 and TLR3 expression were higher than other PRRs in DEF cells suggests that MDA5 and TLR3 are crucial for resisting DTMUV. Subsequently, to explore the functions of TLR3 and MDA5, we over-expressed and silenced TLR3 and MDA5 in the presence or absence of DTMUV infection in DEF cells. Compared with control, important adaptor factors of the TLR3 and MDA5 pathways (e.g., TRIF, IRF7, IPS-1, and NF-κB) were up-regulated in TLR3- and MDA5-over-expressing cells following DTMUV infection. As expected, mRNA levels of downstream inflammatory cytokines (e.g., IL-6, IL-8, and type I IFN) were also increased. Interestingly, DTMUV replication process was specifically impaired when TLR3 or MDA5 were over-expressed. Earlier research demonstrated that type I IFN could impair virus replication, which led us to postulate that type I IFN may repress DTMUV replication in ducks. To clarify the relationship between type I IFN and DTMUV, we co-incubated different concentrations of avian IFN-β with DTMUV infected DEF cells for 12 h. We found that type I IFN exerted a dose-dependent inhibitory effect on DTMUV replication. We then treated DTMUV infected DEF cells with the 500 IU/mL of avian IFN-β for different times and found that avian IFN-β repressed DTMUV replication in a time-dependent manner. These results reveal that DTMUV-infected ducks primarily activate the innate immune signaling pathways of TLR3 and MDA5, and these induce anti-virus responses by producing antiviral factors.

Exosomes are small membrane vesicles (30–150 nm in diameter) enclosed by lipid bilayers [23] and are classified as extracellular vesicles (EVs) [24]. Exosomes originate from the endocytic route and are formed by inward budding of the plasma membrane [25]. Because exosomes encapsulate multiple proteins and nucleic acids such as transforming growth factor-β, TGF-β [26], genomic DNA [27], microRNAs (miRNAs) [28] and so on, they serve crucial roles in intercellular communication [29,30] including regulating innative immunity. Although many exosome functions have been identified in recent years, the functions of DEF-derived exosomes (DEF-Exo) remain unclear. In the present study, we first isolated exosomes derived from DEF cells and explored their functions following DTMUV infection.

Peripheral blood mononuclear cells (PBMCs) are important immune cells that express many TLRs and effectively activate immune responses [31]. Thus, PBMCs are excellent candidates to explore the functions of DEF-Exo. MiRNAs are small (12–23 nucleotides in length), non-coding RNA molecules that can regulate immune responses and exist in exosomes [32]. In the present study, we found that miR-148a-5p is a target miRNA of TLR3 that is significantly decreased both in DEF cells and DEF-Exo following DTMUV infection. To explore the function of miR-148a-5p, we used miR-148a-5p mimic, miR-148a-5p mimic control, DEF-Exo in infected and PBS infected DEF-Exo to co-incubate with PBMCs respectively and examined TLR3 and IFN-β expression. We found miR-148a-5p can negatively regulate TLR3 and IFN-β. We also investigated the effect of TLR3 on miR-148a-5p expression and found that TLR3 can suppress miR-148a-5p expression. Collectively, our results indicate that exosomal miR-148a-5p was significant down-regulated after DTMUV infection and the high level of TLR3 might be one of the reasons, which suggesting that the high level of TLR3 after DTMUV infect can both trigger innate immunity and suppress miR-148a-5p to resist DTMUV.

## 2. Materials and Methods

### 2.1. Ethics Approval and Consent to Participate

The use of duck embryos in this paper was approved by the Animal Ethics Committee of Sichuan Agricultural University (29th 2014, Permit Number: SYXK 2014-187), China. All animal experiments were performed according to the guidelines of the National Institutes of Health and the appropriate biosecurity guidelines. 

### 2.2. Reagents

The antibodies used in this study were as follows: mouse anti-β-actin (66009-1-Ig; Proteintech, Wuhan, China), anti-His (Ht501; TransGen Biotech, Beijing, China), horseradish peroxidase (HRP) goat anti-rabbit IgG antibody (Biodragon Immunotechnologies Co., Ltd., Beijing, China), HRP goat anti-mouse IgG antibody (Biodragon Immunotechnologies Co., Ltd.,.), CD9 (HuaAn Biotechnology Co. Ltd., Hangzhou, China) and mouse anti-tumor susceptibility gene101 (TSG101, Abcam, Cambridge, UK). Avian IFN was purchased from Shanghai Medicine’ nest Pharmaceutical Co.Ltd., and Lipofectamine 3000 was obtained from Invitrogen (Carlsbad, CA, USA). miR-148a-5p mimic and control were purchased from Promega (Madison, WI, USA). 

### 2.3. Cell Lines, Birds, Virus, and Infection

Primary DEF cells were prepared from 9-day-old duck embryos. DEF cells were cultured at 37 °C with 5% CO_2_ in Dulbecco’s Modified Eagle Medium (DMEM) supplemented with 10% fetal bovine serum (FBS). The DTMUV CQW strain (GenBank: KM233707.1) used in this study was obtained from the Key Laboratory of Animal Disease and Human Health of Sichuan Province [33]. Cells were infected with CQW1 DTMUV and incubated for 1 h at 37 °C. Then cells were then washed once with phosphate-buffered saline (PBS) and cultured in DMEM supplemented with 2% FBS at 37 °C with 5% CO_2_ for 12–60 h. Three further passages were done under the same conditions. Then the infected DEF cells were harvested at different time points (12–60 h). Five-day-old specific pathogen-free (SPF) ducks were randomly divided into two groups. The ducks of one group were infected CQW1 DTMUV intramuscularly, and the control group was inoculated in the same manner of sterile phosphate-buffered saline (PBS). Three live ducks, from each group were euthanized at 1, 2, 3, 4, and 5 days post-infection (dpi) and their organs (liver, spleen, and thymus) were collected and stored at −80 °C for further examination.

### 2.4. RNA and cDNA Preparation

Total RNA was extracted from cells and the thymus, liver, and spleen of ducklings using RNAiso plus reagent according to the manufacturer’s instructions (Takara, Otsu, Japan). Agarose gel electrophoresis was used to test RNA sample integrity. The purity of total RNA samples was determined using a Nanodrop 2000 (Thermo Fisher Scientific, Waltham, MA, USA). Then Prime Script™ RT Master Mix (Takara) was used to synthesize the first-strand cDNA. The synthesized cDNA was stored at −20 °C until analysis.

### 2.5. Quantitative Real-Time PCR (qRT-PCR)

Real-Time PCR Detection System (Bio-Rad, Hercules, CA, USA) was used to detect the expression of innate factors according to the manufacturer’s instructions. All reactions were performed in triplicate. Relative expression levels were then calculated by the 2^−ΔΔCT^ method. The primers for DTMUV and the duck IFN-α, IFN-β, MyD88, IL-6, IL-8, IPS-1, NF-κB, TRIF, IRF7, and RIG-I genes were designed using Primer 5 software. The sequences of the other primers used in this study have been described previously (references shown in Table 1). All primers are shown in Table 1. The results were normalized using the housekeeping gene β-actin, and fold changes were determined relative to RNA samples from mock-infected samples. Quantitative real-time PCR was performed in a reaction volume of 10 µL, according to the manufacturer’s instructions. The PCR cycling conditions were: one cycle at 95 °C for 3 min, 39 cycles of denaturation at 95 °C for 10 s and extension for 30 s, followed by a dissociation curve analysis step. Moreover, the transcribing primer of miR-148a-5p was 5′-GTCGTATCCAGTGCAGGGTCCGAGGTATTCGCACTGGATACGACAGTCTGAG-3′, and the qRT-PCR primers of miR-148a-5p were F: 5′-CGGGCAAAGTTCTGTGACACT-3′, R: 5′-CAGTGCAGGGTCCGAGGTAT-3′. These results were normalized using the housekeeping gene U6, and fold changes relative to RNA samples were determined relative to control samples. Quantitative real-time PCR was performed in a reaction volume of 10 μL, according to the manufacturer’s instructions. The PCR cycling conditions were the same as above. 

### 2.6. Recombinant Plasmid Construction

To construct vectors to express TLR3 and MDA5, pairs of specific primers were designed based on the sequences published in GenBank and the pcDNA3.1(+) plasmid from our laboratory. Duck total RNA was extracted using TRIzol, and PrimeScript™ RT Master Mix (Takara, Japan) was used to synthesize the first-strand cDNA. Then duck TLR3 and MDA5 genes were amplified using duck spleen template cDNA. The primers of TLR3 were F: 5′-TGCTCTAGAGCCACCATGGGAAGTGATATTCTTTGT-3′ and R: 5′-TCCGGGCCCTTAGTGGTGATGGTGATGATGTCACCGTGCTTTACTATTAGATTTA AG-3′, the MDA5 primers were F: 5′-TGCTCTAGAGCCACCATGTCGACGGAGTGCCGAGAC-3′, R: 5′-TCCGGGCCCTTAGTGGTGATGGTGATGATGTCAGTCTTCATCACTTGAAGGACAATG-3′. Amplified TLR3 and MDA5 genes were then subcloned into the pcDNA3.1(+) vector between the multiple cloning sites *Apa*I and *Xba*I. Plasmid construction was verified by restriction enzyme digestion. The integrity and fidelity of the inserted fragments were confirmed by sequencing (Wangke, Chengdu, China). The over-expression plasmids were termed pcDNA3.1(+)-TLR3 and pcDNA3.1(+)-MDA5, respectively.

### 2.7. TLR3-shRNA, MDA5-shRNA, and Generation of shRNA-Based Knockdown Cell Lines

The plasmid vector specifically targeting duck MDA5 (MDA5-shRNA), duck TLR3 (TLR3-shRNA) and negative control shRNA (Luc-shRNA) were purchased from GenePharma (Shanghai, China). The TLR3-shRNA sense sequence was: 5′-CACCGCTTGTGTTGTGTGGCAATCATTCAAGAGATGATTGCCACACAACACAAGCTTTTTTG-3′, the antisense sequence was: 5′-GATCCAAAAAAGCTTGTGTTGTGTGGCAATCATCTCTTGAATGATTGCCACACAACACAAGC-3′. The MDA5-shRNA sense sequence was: 5′-CACCGGATGTCGCTACAGAAGATAGTTCAAGAGACTATCTTCTGTAGCGACATCC-3′, the antisense sequence was: 5′-GATCCAAAAAAGGATGTCGCTACAGAAGATAGTCTCTTGAACTATCTTCTGTAGCGACATCC-3′, the NC-shRNA sense sequence was: 5′-CACCGTTCTCCGAACGTGTCACGTTTCAAGAGAACGTGACACGTTCGGAGAATTTTTTG-3′, the Luc-shRNA antisense sequence was: 5′-GATCCAAAAAATTCTCCGAACGTGTCACGTTCTCTTGAAACGTGACACGTTCGGAGAAC-3′. Cell lines stably expressing shRNA specifically targeting MDA5, TLR3, or luciferase control were generated by infection of DEF cells. 

### 2.8. Western Blot Analysis of TLR3, MDA5, and Exosome Proteins

DEF cells were transfected with pcDNA3.1(+)-NC and Luc-shRNA, pcDNA3.1(+)-TLR3 and Luc-shRNA, pcDNA3.1(+)-TLR3 and TLR3-shRNA, pcDNA3.1(+)-MDA5 and Luc-shRNA, or pcDNA3.1(+)-MDA5 and MDA5-shRNA for 24 h and then infected with DTMUV. At 36 h post-infection, total cellular proteins were harvested for transient expression analysis. The transfected cells were re-suspended in 100 μL cell lysis buffer and then stored at −20 °C before analysis.

Samples in loading buffer were boiled for 10 min to eliminate the activity of residual cellular nucleases; however, SNase is thermostable. The samples were then separated by sodium dodecyl sulfate 10% polyacrylamide gel electrophoresis. Western blot analysis was performed using anti-His antibody (1:3000 dilution) and goat polyclonal HRP-conjugated antibodies to mouse IgG (1:3000 dilution). β-actin used as reference protein. Samples were visualized using enhanced chemiluminescence (4A Biotech Co. Ltd., Beijing, China), and the detection of proteins was performed using Quantity One (Bio-Rad).

Exosomes and DEF cell proteins were subjected to electrophoresis and transferred to poly vinylidene fluoride membranes that were probed with the following antibodies: mouse anti-tumor susceptibility gene101 (TSG101,1:3000 dilution), CD9 (1:1000 dilution), and β-actin (1:3000 dilution, internal control). Samples were visualized as described above. 

### 2.9. Exosome isolation

DEF cells were cultured for 24 h, after which the cells were washed once with PBS and cultured in serum-free cell culture medium with or without CQW1 DTMUV for 36 h. Exosomes were purified from serum-free supernatants of DEF cells with or without DTMUV infection according to the manufacturer’s instructions using VEX exosome Isolate Reagent (Vazyme, Guilin, China). Purified exosomes were dissolved in PBS and visualized by electron microscopy and other techniques.

### 2.10. PBMC Preparation

PBMCs were isolated from peripheral blood samples using the duck peripheral blood leukocyte separation solution kit (TBD science, Tianjin, China) according to the manufacturer’s instructions.

### 2.11. Interaction of Exosomes with PBMCs

DEF-Exo were labeled with the green lipophilic fluorescent dye PKH67 using the Green Fluorescent Cell Linker Midi Kit (Sigma-Aldrich, St. Louis, MO, USA) according to the manufacturer’s instructions, and incubated with PBMCs. After incubation for 3, 6, or 9 h, cells were immobilized with 4% paraformaldehyde, and the nuclei were stained with DAPI. The cells were then visualized by fluorescence microscopy.

### 2.12. Enzyme-Linked Immunosorbent Assay (ELISA) to Detect Duck TLR3 Protein

PBMCs were prepared as above described. DEF-derived exosomes that untreated DTMUV and DEF-derived exosomes infected with DTMUV, miR-148a-5p mimic, or control were co-incubated with PBMCs respectively for 24 h. Three further passages of the cells were performed. Then, culture supernatants were tested for the presence of duck TLR3 protein using an ELISA kit (MEIMIAN, China), following the manufacturer’s introductions.

### 2.13. Dual-Luciferase Reporter Assay

By comparing sequences, we found that miR-148a-5p could target the CDS region of TLR3. Thus a 400 bp fragment of the TLR3 coding region (CDS) including wild-type (wt) or mutant (mt) miR-148a-5p binding sites was cloned downstream of the firefly luciferase gene in the PmiRGLO vector (Promega, Madison, WI, USA). The resulting plasmids were named TLR3-wt and TLR3-mt, respectively. For reporter assays, DEF cells were first co-transfected with TLR3-wt or TLR3-mt with miR-148a-5p mimic using Lipofectamine 3000 reagent (Invitrogen), while both miR-148a-5p and no-load PmiRGLO vector were transfected as controls. Luciferase activity was measured using the Dual-Luciferase Reporter Assay System (Promega).

### 2.14. Statistical Analysis

All results are shown as mean ± standard error (SE). Statistical significance was determined by Student’s *t*-tests. *P* < 0.05 was considered significant.

## 3. Results

### 3.1. DTMUV-Infected Ducks Show Up-Regulated the Expression of TLR3, MDA5, and Inflammatory Cytokines

Based on a previous study [21], DTMUV infection of CEF cells and HEK293 cells can trigger a robust innate immune response, in which TLR3 and MDA5 play an important role. Moreover, Yu et al. [22] performed high-throughput sequencing and found that DTMUV-infected DEF cells exhibit various innate immunity processes, which notably include the TLRs and RLRs signaling pathways. Moreover, we found other flaviviruses, like dengue virus (DENV), Zika virus (ZIKV) and West Nile virus (WNV) can also trigger robust innate immunity processes including TLRs and RLRs signaling pathways. Therefore, we speculate DTMUV infected ducks would also elicit similar innate immunity processes. To determine whether DTMUV infection of ducklings can also elicit robust innate immunity, we examined the expression of TLRs, RLRs, type I IFN, and pro-inflammatory cytokines in thymus, liver, and spleen tissues. First, 5-day-old ducklings were challenged with DTMUV or PBS, after which thymus, liver, and spleen samples were harvested during the early post-infection period. In general, the expression of all cytokines was increased, but the degree of the increase differed depending on tissue type and time. In thymus tissue, IFN-α and IFN-β both peak on day 2 after infection (Figure 1A,B). Pro-inflammatory cytokines IL-6 and IL-8 peak on the 3 dpi and 2 dpi, respectively (Figure 1C,D). In addition, important immune factors in the RIG-I and TLRs pathways are also significantly increased after DTMUV infection. Among them, MDA5 peaked on the 3 dpi (Figure 2A), MyD88 peaked on the 4 dpi (Figure 2B), and TLR2 peaked on the 2 dpi (Figure 2C).

In liver tissues, IFN-α and IFN-β expression trends were different from those in the thymus. Although the expression levels were also significantly increased, in liver tissues, IFN-α expression levels were significantly increased in the first two days, then the expression trend was stable (Figure 3A). IFN-β peaked 4 dpi (Figure 3B). The expression of pro-inflammatory factor IL-6 peaked on the 4 dpi (Figure 3C). However, the expression trend of IL-8 was somewhat complicated, reaching a peak on the first day, then decreased compared with the first day, and increased again on the 4 dpi (Figure 3D). The expression level of the major immune factor MDA5 in the RIG-I pathway peaked on 2 dpi (Figure 4A), and the major immune factors in the TLRs pathway MyD88, TLR2 and TLR4 peaked on 3 dpi, 2 dpi, and 3 dpi respectively (Figure 4B–D).

In spleen tissues, although the expression level of IFN-α is significantly increased, its expression tends to decrease with time (Figure 5A). The expression trend of pro-inflammatory factor IL-6 was more complex, peaking on the first day after infection, then decreasing, and increasing again on the 4 dpi (Figure 5B). The expression trend of IL-8 is also relatively complex, decreasing after increasing on the first day and increasing again on the 3 dpi (Figure 5C). The expression level of the major immune factor MDA5 in the RIG-I pathway peaked on the 3 dpi (Figure 5D), and the major immune factors in the TLRs pathway MyD88, TLR2, and TLR3 peaked on the 2 dpi, 3 dpi, and the 1 dpi respectively (Figure 6A–C). These results provide strong evidence that duck innate immune responses can be triggered by DTMUV infection in vivo.

### 3.2. TLR3 and MDA5 Play Major Immunity Roles in DTMUV-Infected DEF Cells

Although Yu et al. [22] reported expression changes for some innate immune cytokines in DTMUV-infected DEF cells, the degrees of change and trends of expression were not described. To address this, we infected DEF cells with DTMUV and assessed the expression of TLRs, RLRs, inflammatory cytokines and type I IFN at different time points. Similar to the results for DTMUV-infected ducklings, MDA5, and TLR3 expression were significantly increased in the early post-infection period compared with other PRRs. Specifically, the expression levels of IFN-α and IFN-β increased with the time of virus infection, in which IFN-α peaked at 48 h, while the expression levels of IFN-β did not decline within 60 h after infection (Figure 7A,B). Pro-inflammatory factor IL-6 did not decline within 60 h after infection, while IL-8 expressions peaked at 48 h (Figure 7C,D). The expression levels of both RIG-I and MDA5, important factors in the RIG-I pathway, were also increased, in which MDA5 peaked at 48 h after infection, while RIG-I peaked at 36 h, then decreased, and increased again at 60 h (Figure 7E and Figure 8D). However, the expression trend of immune factors in the TLRs pathway is relatively simple. MyD88, TLR2, TLR3, and TLR4 peak at 48, 24, 12, and 36 h after infection (Figure 7F and Figure 8A–C). Then we compared the expressions of RLRs and TLRs after infection, and found that the expressions of MDA5 were much higher than RIG-I (Figure 8E), while the expressions of TLR3 was much higher than those of TLR2 and TLR4 (Figure 8F). These results demonstrate that DTMUV can trigger a strong innate immune response in vitro, in which TLR3 and MDA5 appear to have important roles.

### 3.3. MDA5 and TLR3-Dependent Signaling Pathways have Important Roles in Immunity During DTMUV Infection

The above study showed that MDA5 and TLR3 are significantly increased in vivo and in vitro during DTMV infection. Therefore, we next investigated the functions of TLR3 and MDA5 in innate immunity. To explore the roles of TLR3 and MDA5, we over-expressed or silenced MDA5 or TLR3 in DEF cells. We then detected cytokine expression with qRT-PCR. To over-express TLR3 and MDA5, eukaryotic expression vectors containing TLR3 or MDA5 CDs were successfully constructed and named pcDNA3.1(+)-TLR3 and pcDNA3.1(+)-MDA5, respectively. Since we cannot find the duck antibodies of TLR3 and MDA5 either on the market or in our laboratory, we added His tags to the plasmids to detect the expression of TLR3 and MDA5 by anti-His antibody. To silence TLR3 and MDA5, specific shRNAs were designed and synthesized by GenePharma, which were named TLR3-shRNA and MDA5-shRNA. We generated pcDNA3.1(+)-NC and Luc-shRNA as negative controls. For over-expression of TLR3 and MDA5, we co-transfected pcDNA3.1(+)-TLR3/Luc-shRNA and pcDNA3.1(+)-MDA5/Luc-shRNA into DEF cells, respectively. pcDNA3.1(+)-TLR3/TLR3-shRNA and pcDNA3.1(+)-MDA5/MDA5-shRNA were transfected into DEF cells to inhibit TLR3 and MDA5 expression respectively. After 24 h, DTMUV was added to the cultures, and cells were harvested at 36 hpi. To confirm that MDA5 and TLR3 were successfully over-expressed or knocked down, we detected the mRNA and protein levels by qRT-PCR and Western blot analysis, respectively. As shown in Figure 9A,B, TLR3 and MDA5 were successfully over-expressed or knocked down at both the mRNA and protein levels. We next examined the mRNA levels of inflammatory cytokines, type I IFN and signal adaptors by qRT-PCR in the over-expressing and silenced lines. As shown in Figure 9C and Figure 10A, disruption of TLR3 or MDA5 expression resulted in low mRNA levels of IL-6, IL-8, and IFN-β. In contrast, over-expression of TLR3 and MDA5 increased the levels of these cytokines and type I IFN. Moreover, the expression of important adaptors of TLR3 (TRIF, IRF7, and NF-κB) and MDA5 (IPS-1, NF-κB) signaling pathways were also enhanced (Figure 9D and Figure 10B). However, disrupting TLR3 or MDA5 expression impaired the expression of these adaptors. Collectively, these results suggest that TLR3 and MDA5 are important innative factors to resist DTMUV, in which the TLR3 and MAD5 signaling can both target NF-κB to induce abundant inflammatory cytokines and type I IFN. Additionally, the TLR3 signaling pathway can target IRF7 to induce type I IFN.

### 3.4. Type I IFN Significantly Impairs DTMUV Replication in Dose- and Time-Dependent Manners

To investigate whether TLR3 and MDA5 can influence DTMUV replication, we over-expressed and silenced TLR3 or MDA5 in DEF cells. DTMUV envelop gene expression was then detected by qRT-PCR. As expected, TLR3 and MDA5 over-expression were found to impair DTMUV replication (Figure 10C,D). Many previous reports showed that type I IFN can inhibit the replication of viruses including dengue [39] and DTMUV [21]. We wondered whether type I IFN has antiviral activity in response to DTMUV infection in ducks. To address this, DEF cells were infected with DTMUV for 36 h and then treated with avian IFN-β at concentrations of 50, 100, 250, and 500 IU/mL. We found that avian IFN-β impaired DTMUV replication process in a dose-dependent manner (Figure 11A). Moreover, to determine whether the observed antiviral effects of avian IFN-β were time-dependent, we treated DEF cells with 500 IU/mL avian IFN-β for 0, 6, 12, or 24 h after infecting with DTMUV for 36 h. We found that the expression of E protein decreased with time (Figure 11B). These results suggest that MDA5 and TLR3 are important for inducing type I IFN and triggering effective antiviral immunity.

### 3.5. Morphological Characterization and Identification of Exosomes

Exosomes are small membrane vesicles (30–150nm in diameter) that contain various substances including miRNAs. So far, many researches have explored that exosome-miRNA can regulate immune pathways via regulating the expression of immune factors [40,41]. Therefore, in this study we wonder whether there has a specific Exo-miRNA can regulate the expression of TLR3 or MDA5 to influence the antiviral processes after DTMUV infection. To address this, we first isolated exosomes and confirmed their morphological features by transmission electron microscopy. As shown in Figure 12A, exosome protein markers, CD9 and TSG101 were identified by Western blot analysis both in exosomes and in DEF cells. Figure 12B shows DEF-Exo samples contained nanometer-sized microvesicles.

### 3.6. DTMUV-Infected DEF Cells and DEF-Derived Exosomes Show Decreased Levels of miR-148a-5p

Previous reports have shown that miRNAs can regulate TLRs pathways in many diseases [40,41]. As such, we wondered whether miRNAs could regulate TLR3 or MDA5 expression in duck. In this study, we found that miR-148a-5p is a potential target miRNA of TLR3 by using miRanda and RNAhybrido. Unfortunately, we did not find miRNA that can potential target MDA5, thus in our study, we only explored the relationship between miR-148a-5p and TLR3. Firstly, we examined miR-148a-5p expression in DEF cells and DEF-derived exosomes with or without DTMUV infection by qRT-PCR. According to the results, we found that miR-148a-5p expression was reduced in both DEF cells and DEF-derived exosomes upon DTMUV infection compared with the non-infected controls (Figure 13A). 

### 3.7. miR-148a-5p Targets TLR3

After comparing the sequences of miR-148a-5p and TLR3, we found putative miR-148a-5p binding sites in the coding region of TLR3 (Figure 13B). To verify the prediction, we transfected DEF cells with a miR-148a-5p mimic or control miRNA together with the reporter vector. The reporter vectors contained the wild type copy of the miR-148a-5p seed-binding site (TLR3-wt) or a mutated coding region sequence of TLR3 (TLR3-mt). At 48 h after transfection, we performed dual-luciferase reporter assays on DEF cells. As shown in Figure 13C, the miR-148a-5p mimic significantly decreased luciferase activity compared to control miRNA, while TLR3-mt led to complete abrogation of the negative effect. These results suggest that miR-148a-5p can directly target TLR3.

### 3.8. miR-148a-5p and Exo-miR-148a-5p Negatively Regulate TLR3 and IFN-β Expression in PBMCs

One important function of exosome is to pass information between cells. Therefore, to explore the function of miR-148a-5p, we treated PBMCs with miR-148a-5p mimic and miRNA control respectively, when we considered PBMC is an important immune cell and can play an important immune role. According to our results, we observed low mRNA and protein levels of TLR3 when PBMCs were treated with miR-148a-5p mimic (Figure 14A,B), meaning miR-148a-5p can negatively regulate the expression of TLR3. Moreover, we found the level of IFN-β mRNA also decreased (Figure 14C). We next explored the biological function of Exo-miR-148a-5p. First, we demonstrated that PBMCs could ingest DEF-Exo by fluorescence microscopy (Figure 15). We next examined the protein level of TLR3 and mRNA levels of TLR3 and IFN-β after we co-incubate DTMUV infected DEF-Exo (contain relatively less miR-148a-5p, exosome-infected) or PBS infected DEF-Exo (contain relatively more miR-148a-5p, exosome-control) with PBMCs. As shown in Figure 14A–C, compared with exosome-infected, exosome-control appeared to have reduced expression of TLR3 (at both mRNA and protein levels) and IFN-β (mRNA level). This is consistent with our results that miR-148a-5p could negatively regulate the expression of TLR3.

### 3.9. TLR3 May Be One of the Reasons for the Decrease of miR-148a-5p 

To further explore the relationship between TLR3 and miR-148a-5p, we examined the expression of miR-148a-5p after over-expressing or silencing TLR3 in DEF cells. miR-148a-5p expression was significantly decreased following TLR3 over-expression compared with the silencing of TLR3. These results suggest that TLR3 and miR-148a-5p expression are oppositely regulated (Figure 14D). In addition, the negative regulation of miR-148a-5p by TLR3 may be one of the reasons for the low expression of miR-148a-5p after DTMUV infection, but more experiments are required to explore the mechanism by how TLR3 negatively regulates miR-148a-5p.

## 4. Discussion

DTMUV causes serious disease in waterfowl, which has become increasingly widespread from southeast to north China and has resulted in considerable economic losses [1]. While morbidity is somewhat low in adult animals, it can be much higher in infected ducklings [2,3]. Unfortunately, there is no effective way to prevent DTMUV. TLRs and RLRs are important mediators of innate immunity, and play essential roles in resisting invasion by several viruses including flavivirus-like dengue virus (DENV), West Nile virus (WNV), and DTMUV. According to previous studies, TLR3 is involved in resisting DENV [42,43], WNV [44], and DTMUV [21], while MDA5 is involved in defense against DENV [45] and DTMUV [21]. Our experiments were aimed at exploring the specific mechanism underlying innate immunity in ducks infected with DTMUV. Firstly, we found compared with other innate immune factors, the expression levels of TLR3 and MDA5 were significantly increased through qRT-PCR indicating that TLR3 and MDA5 play an important role in the anti-DTMUV process. Then we explored the immune function of TLR3 and MDA5 through over-expressing or silencing TLR3 and MDA5 respectively. According to our results TLR3 and MDA5 are crucial to induce INF-β, which is an important anti-virus cytokine. In our view, these results may be one of the theoretical bases for developing flavivirus vaccines or treatments. Moreover, our results showed that the inhibitory effect of IFN-β on DTMUV was time-dependent and dose-dependent, thus IFN-β could be used as one of the treatment methods for DTMUV. However, exactly what dose of IFN-β has the best antiviral effect and how long the antiviral effect last requiring further testing. Although we found that TLR3 and MDA5 can promote the expression of IFN-β, it can be seen from Figure 7 and Figure 8 that the expression trend of TLR3, MDA5 and IFN-β is different. When the expression trend of TLR3 and MDA5 begins to decline, the expression of IFN-β is still increasing, indicating that TLR3 and MDA5 are important for inducing IFN-β, but they are not the only immune factors to promote the expression of IFN-β. Actually, except TLR3 and MDA5, Chen et al. reveals that stimulator of interferon gene (STING) is also vital for duck type I interferon induction [46]. 

Exosomes are critical components of cellular communication that contain various materials including miRNAs [28]. Many studies have demonstrated the influence of exosomes on innate immunity. Exosomal miRNA-146a was found to represses TLR2 expression in Alzheimer’s disease [41], exosomes derived from pancreatic cancer contain high levels of miRNA-203, which down-regulates TLR4 and cytokines in DCs [47], exosomes abundant in miR-124 in amyotrophic lateral sclerosis are associated with persistent NF-κB activation and up-regulated expression of inflammatory factors including TLR4 [40]. Based on this knowledge, in the present study, we explored how DEF-Exo miRNA regulates innate immunity in ducks. According to our results, we found that miR-148a-5p can target and down-regulate the expression of TLR3 and IFN-β, which is a negative factor during DTMUV infection. However, our following results show that after DTMUV infects, the expression of DEF-Exo miR-148a-5p is significantly decreased, and the high level of TLR3 is one of the reasons. All of these results suggest that after DTMUV infects, the high level of TLR3 can both increase the expression of IFN-β and decrease the level of miR-148a-5p to resist DTMUV infection. In addition, although we have confirmed that TLR3 can negatively regulate miR-148a-5p, the mechanism by how TLR3 regulates miR-148a-5p expression remains to be explored.

Although our work represents the first demonstration of successful isolation of duck exosomes and allowed us to explore exosomal miRNA functions, our study has some limitations. Firstly, miRNA silencing is easily accomplished because only 7–8 nt of a target gene is needed to achieve the effect. Thus, there might be other miRNA-148a-5p target genes involved in regulating duck immune processes that we may have missed. Actually, according to Min C et al. study [38], there are 48 miRNAs, including 26 known miRNAs and 22 novel miRNAs were differentially expressed in response to DTMUV infection, thus more researches are required to explore how miRNAs to regulate antiviral processes after DTMUV infection. Unfortunately, this microRNA expression profiles can only be used during DTMUV infection because different viruses will induce different microRNA expression profiles, and even different species of ducks might express different microRNA expression profiles, thus more experiments are required. Secondly, research into exosome function is still in preliminary stages. Recent reports revealed that not only the substances that packaged by exosomes [48] but also surface proteins [49] have functions to regulate innate immune responses. However, our research only explored the function of miR-148a-5p in duck during DTMUV infection. To further investigating how DTMUV interacts with ducks and how exosomes regulate duck antiviral processes, it is a good start to detect what materials DEF-Exo contains and how these materials change after DTMUV infection. Thirdly, our study was limited to cultured cells. The functions of DEF-Exo in vivo remain to be elucidated. Although the technology to explore the biological function of exosomes in vitro is not widely available, some studies have injected isolated exosomes into animals to track their locations and study their functions. Additional studies are needed to illuminate the biological functions of DEF-Exo. Lastly, many studies have revealed the mechanism of DTMUV-host interactions, and the non-structural proteins NS1, NS2B3, and NS3 are involved in regulating immunity to resist DTMUV or help DTMUV escape host immune responses. NS1 inhibits IFN-β expression by inhibiting the activity of RIG-I [50], NS2B3 inhibited RIG-I, MDA5, MAVS, and STING directed IFN-β transcription [51], NS3 can regulate p38/mitogen-activated protein kinase pathway (p38/MAPK) to avert OS, causing apoptosis, making it possible for viruses to escape host immune responses [52]. Although our experiments indicating that TLR3 and MDA5 has important immunological effects against viruses, how DTMUV interact with host immunity have not been elucidated. As we can learn from previous studies, virus-host interactions are complex. On the one hand, the virus can trigger the immune process; on the other hand, the virus can escape host immune responses. Therefore, a more thorough understanding of DTMUV interaction with the host is important for the prevention and treatment of DTMUV, and our results provide basic information.

## 5. Conclusions

Our experiment results show that DTMUV can trigger duck robust innate immune process, in which TLR3 and MDA5 have important immunological functions, they affect DTMUV replication by promoting IFN-β expression. Then we successfully isolated exosomes secreted by DEF cells and found the expression of miR-148a-5p in exosome is significantly decreased following DTMUV infect, and the high level of TLR3 might one of the reasons. Our results also show that miR-148a-5p can target TLR3 and decrease the expression of TLR3 and IFN-β, negatively regulate innate immunity. The same result is observed when we co-culture DEF-Exo induced from DTMUV infected DEF cells (relatively has low level of miR-148a-5p), DEF-Exo induced from PBS control group (relatively has high level of miR-148a-5p) with PBMC, These results suggest that the high level of TLR3 after DTMUV can both increase the expression of IFN-β and decrease the level of miR-148a-5p to resist DTMUV infection.

## Figures and Tables

**Figure 1 viruses-12-00094-f001:**
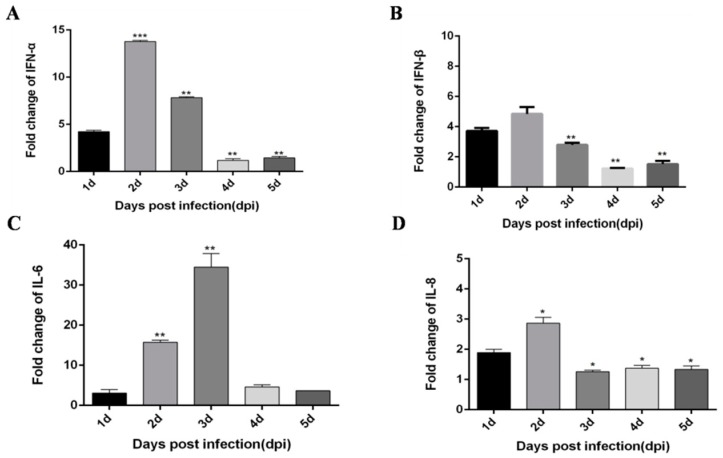
Duck tembusu virus (DTMUV)-infected ducks effectively up-regulate IFN-α, IFN-β, IL-6, IL-8 expression in the thymus. Each young specific pathogen-free (SPF) duck was challenged by intramuscular inoculation with DTMUV. (**A**–**D**) Thymus tissues of mock- and DTMUV-infected ducklings were collected at the indicated time points for examination of IFN-α (**A**), IFN-β (**B**), IL-6 (**C**), and IL-8 (**D**) mRNA expression using qRT-PCR. The average levels from three independent experiments are plotted. The error bars represent the SEs. * *P* < 0.05, ** *P* < 0.01.

**Figure 2 viruses-12-00094-f002:**
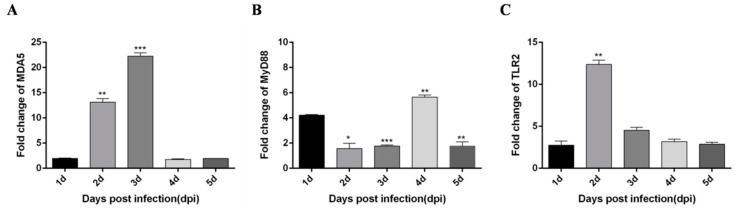
DTMUV-infected ducks effectively up-regulate MDA5, MyD88, TLR2 expression in the thymus. Each young SPF duck was challenged by intramuscular inoculation with DTMUV. (**A**–**C**) Thymus tissues of mock- and DTMUV-infected ducklings were collected at the indicated time points for examination of MDA5 (**A**), MyD88 (**B**), and TLR2 (**C**) mRNA expression using qRT-PCR. The average levels from three independent experiments are plotted. The error bars represent the SEs. * *P* < 0.05, ** *P* < 0.01.

**Figure 3 viruses-12-00094-f003:**
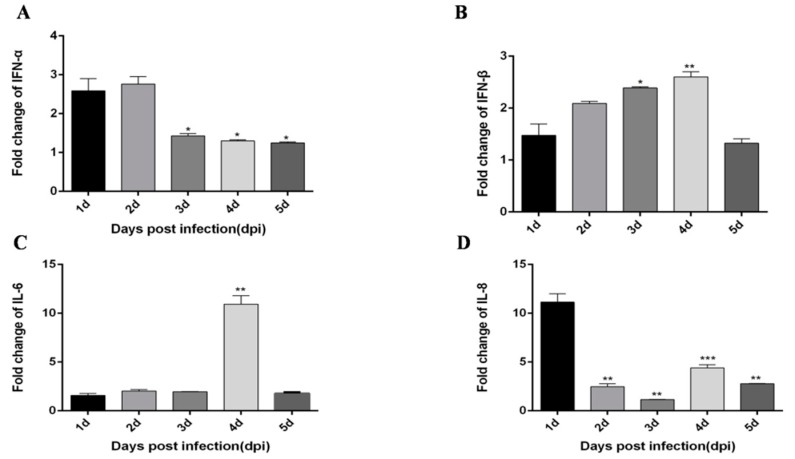
DTMUV-infected ducks effectively up-regulate IFN-α, IFN-β, IL-6, IL-8 expression in the liver. Each young SPF duck was challenged by intramuscular inoculation with DTMUV. (**A**–**D**) Liver tissues of mock- and DTMUV-infected ducklings were collected at the indicated time points for examination of IFN-α (**A**), IFN-β (**B**), IL-6 (**C**), IL-8 (**D**) mRNA expression using qRT-PCR. The average levels from three independent experiments are plotted. The error bars represent the SEs. * *P* < 0.05, ** *P* < 0.01.

**Figure 4 viruses-12-00094-f004:**
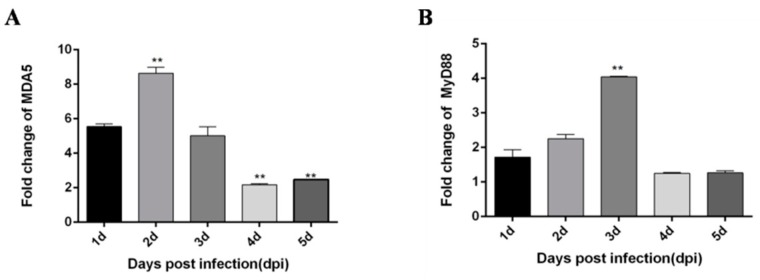
DTMUV-infected ducks effectively up-regulate MDA5, MD88, TLR2, and TLR4 expression in the liver. Each young SPF duck was challenged by intramuscular inoculation with DTMUV. (**A**–**D**) Liver tissues of mock- and DTMUV-infected ducklings were collected at the indicated time points for examination of MDA5 (**A**), MyD88 (**B**), TLR2 (**C**), and TLR4 (**D**) mRNA expression using qRT-PCR. The average levels from three independent experiments are plotted. The error bars represent the SEs. * *P* < 0.05, ** *P* < 0.01.

**Figure 5 viruses-12-00094-f005:**
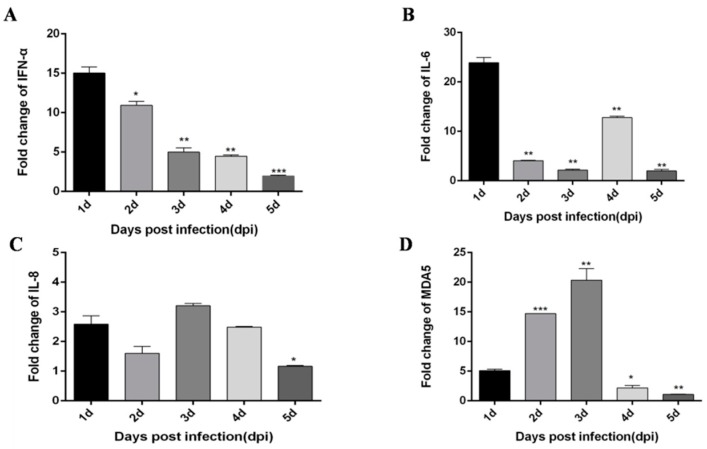
DTMUV-infected ducks effectively up-regulate IFN-α, IL-6, IL-8, MDA5 expression in the spleen. Each young SPF duck was challenged by intramuscular inoculation with DTMUV. (**A**–**D**) Spleen tissues of mock- and DTMUV-infected ducklings were collected at the indicated time points for examination of IFN-α (**A**), IL-6 (**B**), IL-8 (**C**), and MDA5 (**D**) mRNA expression using qRT-PCR. The average levels from three independent experiments are plotted. The error bars represent the SEs. * *P* < 0.05, ** *P* < 0.01.

**Figure 6 viruses-12-00094-f006:**
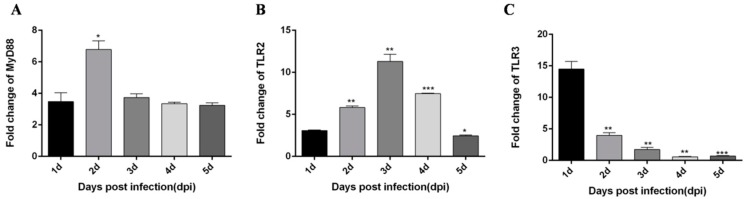
DTMUV-infected ducks effectively up-regulate MyD88, TLR2, TLR3 expression in the spleen. Each young SPF duck was challenged by intramuscular inoculation with DTMUV. (**A**–**C**) Spleen tissues of mock- and DTMUV-infected ducklings were collected at the indicated time points for examination of MyD88 (**A**), TLR2 (**B**), and TLR3 (**C**) mRNA expression using qRT-PCR. The average levels from three independent experiments are plotted. The error bars represent the SEs. * *P* < 0.05, ** *P* < 0.01.

**Figure 7 viruses-12-00094-f007:**
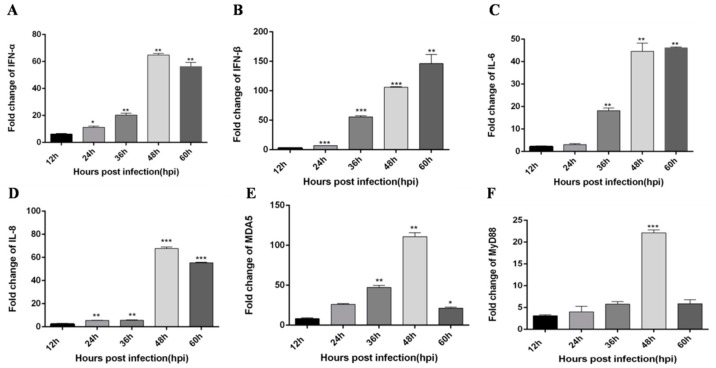
DTMUV-infected ducks effectively up-regulate IFN-α, IFN-β, IL-6, IL-8, MDA5, and MyD88 expression in DTMUV-infected DEF cells. DEF cells were infected with or without DTMUV and harvested at 12, 24, 36, 48, and 60 hpi. The qRT-PCR analysis was performed to examine the mRNA expression levels of type I IFN (**A**,**B**), (RIG-I)-like receptors (RLRs) (**E**), MyD88 (**F**), IL-6 (**C**), and IL-8 (**D**). The average levels from three independent experiments are plotted. The error bars represent the SEs. * *P* < 0.05, ** *P* < 0.01.

**Figure 8 viruses-12-00094-f008:**
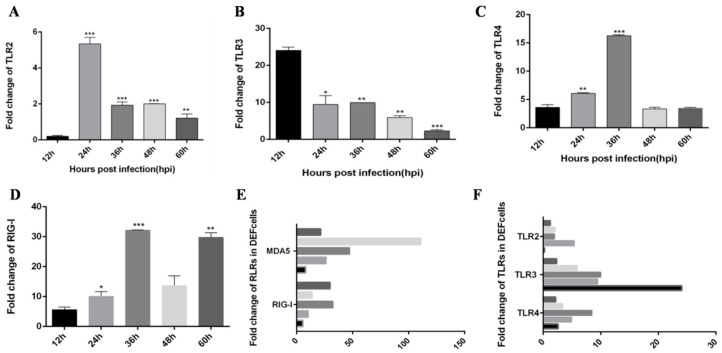
TLR3 and MDA5 play important immunity roles in DTMUV-infected DEF cells. DEF cells were infected with or without DTMUV and harvested at 12, 24, 36, 48, and 60 hpi. The qRT-PCR analysis was performed to examine the mRNA expression levels of RLRs (**D**), and Toll-like receptors (TLRs) (**A**–**C**). Comparison of TLRs and RLRs in DEF cells (**E**,**F**). The average levels from three independent experiments are plotted. The error bars represent the SEs. * *P* < 0.05, ** *P* < 0.01.

**Figure 9 viruses-12-00094-f009:**
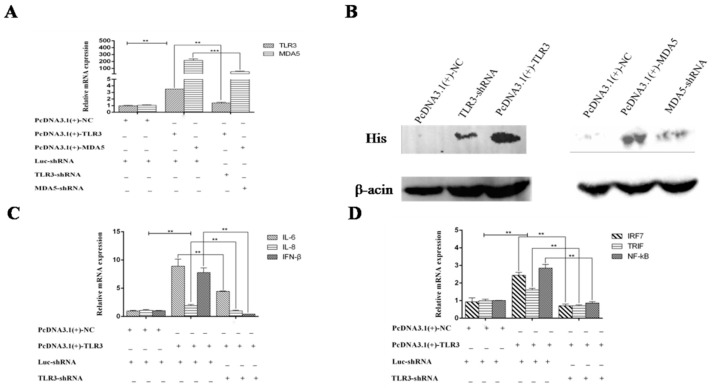
MDA5- and TLR3-dependent signaling pathways play important roles in immunity during DTMUV. DEF cells stably expressing shRNA specifically targeting either TLR3, MDA5, or luciferase control were infected with DTMUV for 36 h. Quantitative real-time PCR and Western blot analysis were then performed to determine interference efficiency for TLR3 and MDA5 (**A**,**B**). The expression of IL-6, IL-8, IFN-β, IRF7, TRIF, and NF-κB in DEF cells stably expressing shRNA targeting TLR3 and pcDNA3.1(+)-TLR3 were analyzed by quantitative real-time PCR (**C**,**D**). In (**A**,**C**,**D**) we transfected pcDNA3.1(+)-nc and Luc-shRNA as control group, and set them as 1. The average levels from three independent experiments are plotted. The error bars represent the S.E. * *P* < 0.05, ** *P* < 0.01.

**Figure 10 viruses-12-00094-f010:**
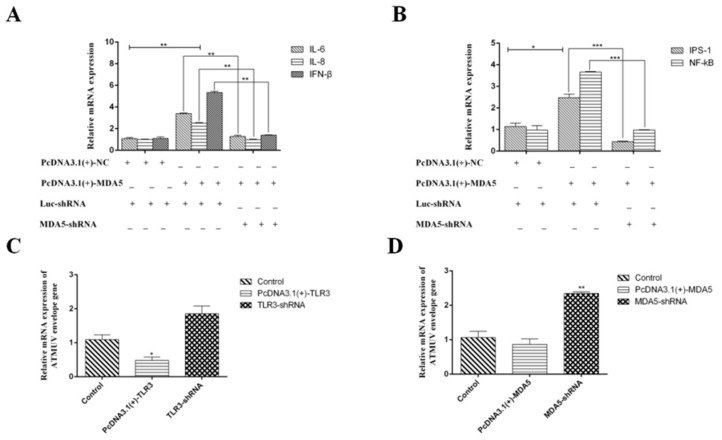
MDA5- and TLR3-dependent signaling pathways play important roles in immunity during DTMUV infection. The expression of IL-6, IL-8, IFN-β, IPS-1 and NF-κB expression in DEF cells stably expressing shRNA targeting MDA5 and pcDNA3.1(+)-MDA5 were also analyzed by quantitative real-time PCR (**A**,**B**). (**C**,**D**) shows the expression of the DTMUV E protein after silencing or over-expressing TLR3 and MDA5 respectively. The average levels from three independent experiments are plotted. In (**A**,**B**) we transfected pcDNA3.1(+) -nc and Luc-shRNA as control group, and set them as 1. In (**C**,**D**) we set control group as 1. The error bars represent the S.E. * *P* < 0.05, ** *P* < 0.01.

**Figure 11 viruses-12-00094-f011:**
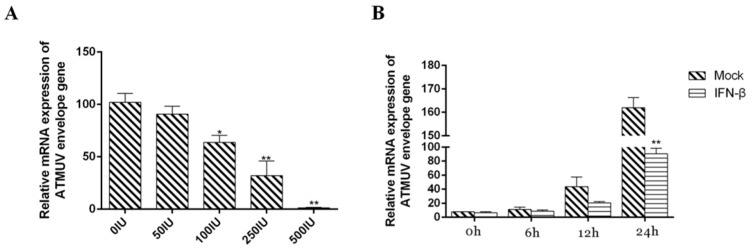
IFN significantly impairs DTMUV replication in dose- and time-dependent manners. (**A**) DEF cells were incubated with avian IFN-β ranging from 50 to 500 IU/mL after infection with DTMUV for 36 h. qRT-PCR was performed to examine DTMUV envelope gene expression. (**B**) DEF cells were incubated with avian IFN-β (500IU/mL) for 0, 6, 12, or 24 h following infection with DTMUV at for 36 h. qRT-PCR was performed to examine DTMUV envelope gene expression. The error bars represent the SEs. * *P* < 0.05, ** *P* < 0.01.

**Figure 12 viruses-12-00094-f012:**
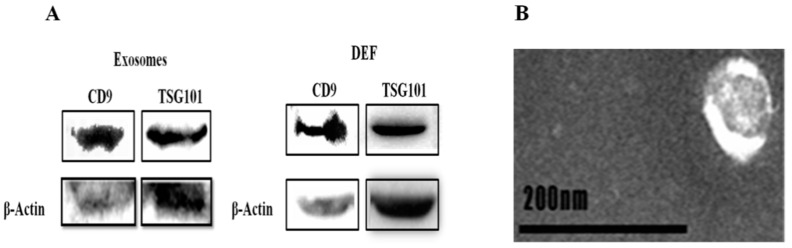
Morphological characterization and identification of exosomes. Exosomes were purified from serum-free medium of DEF cells, exosome protein markers (CD9 and TSG101) were identified by Western blot analysis. β-actin was used as an internal control. (**A**) their morphological features were observed by electron microscopy. Exosomes were nanometer-sized microvesicles (20–120 nm) (**B**).

**Figure 13 viruses-12-00094-f013:**
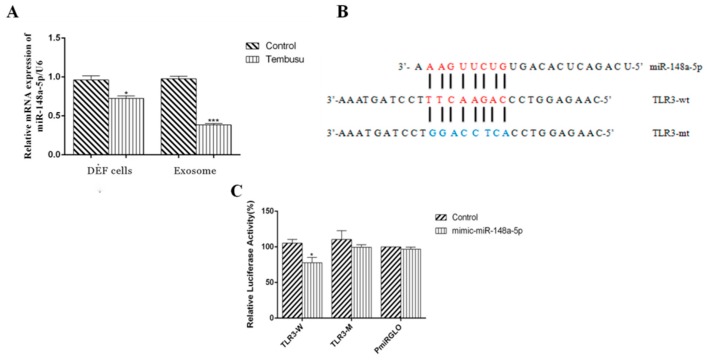
miR-148a-5p directly targets the TLR3 coding region in DEF cells. (**A**) miR-148a-5p expression in DEF cells and DEF-Exo is determined by qRT-PCR. U6 snRNA was used as an internal control. The average levels from three independent experiments are plotted. (**B**) is the TLR3 coding region containing the reporter constructs. (**C**) Luciferase reporter assays in DEF cells upon co-transfection of the WT (TLR3-W) or mutant (TLR3-M) TLR3 coding region with miR-148a-5p mimic and control. In (**A**) we set the control group as 1, and in (**C**) we set the control group as 100%. These experiments were performed in triplicate, and the results are shown as mean ± SEs * *P* < 0.05, ** *P* < 0.01.

**Figure 14 viruses-12-00094-f014:**
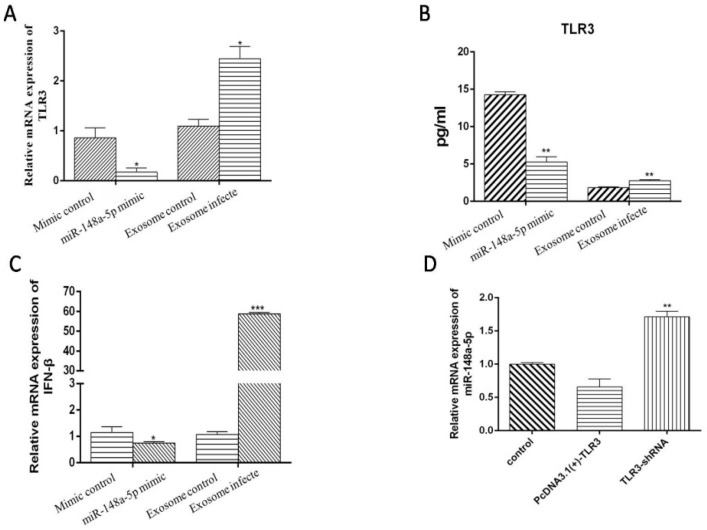
miR-148a-5p and Exo-miR-148a-5p negatively regulates TLR3 and IFN-β expression in PBMCs. After treating PBMCs with miR-148a-5p mimic and control, the mRNA and protein expression of TLR3 decreased (**A**,**B**), as did the mRNA level of IFN-β (**C**). After treatment of PBMCs with DEF-derived exosomes with or without DTMUV infection, the mRNA and protein expression of TLR3 decreased (**A**,**B**), as did the mRNA level of IFN-β (**C**). (**D**) DEF cells were treated with pcDNA3.1(+)-NC, pcDNA3.1(+)-TLR3, and TLR3-shRNA for 36 h, after which miRNA-148a-5p expression was determined by qRT-PCR. The results show that TLR3 can suppress miR-148a-5p expression. In (**C**) we set mimic control and exosome control as 1, and in (**D**) we set the control as 1. The error bars represent the SEs. * *P* < 0.05, ** *P* < 0.01.

**Figure 15 viruses-12-00094-f015:**
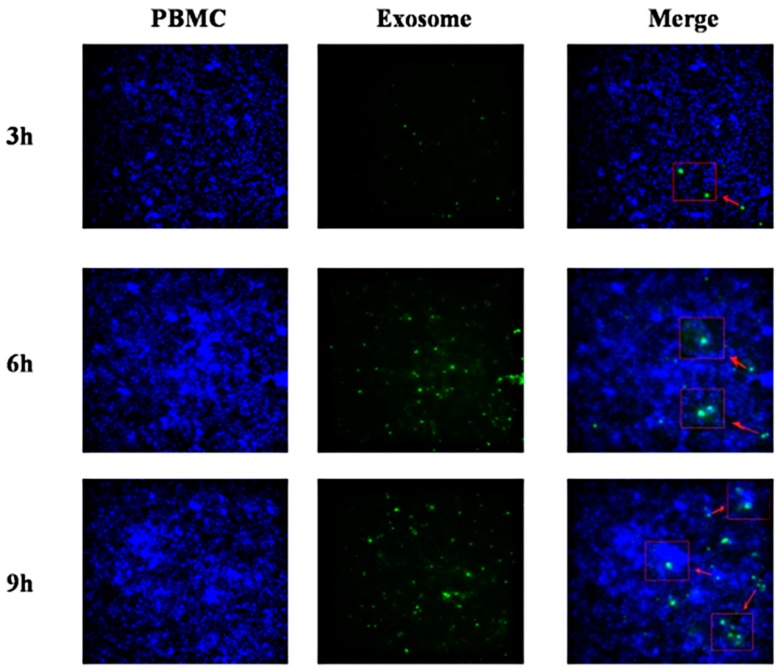
PBMCs ingest DEF-exosomes. To demonstrate that PBMCs could ingest exosomes secreted from DEF cells, we first stained exosome with PKH67 green fluorescent dye and co-cultured with PBMCs for 3, 6, and 9 h, and then the nucleus of PBMC were stained. Lately, we used fluorescence microscopy to detect the PBMC nucleus and exosomes respectively and then merged these pictures. From the merged pictures (200 μm), we can see that exosomes clustered around the nucleus of the PBMC, suggesting that exosomes were ingested by the PBMC.

**Table 1 viruses-12-00094-t001:** Primers used in this study.

Primer Name	Oligonucleotide Sequence (5′→3′)	Length	References
TLR2	F1: TCCTTCATTCAGCACCAGGC R2: GAAAAACACAGCGCAGATCA	171 bp	[34]
TLR3	F2: ATGTCATGCAAACCTGACCA R2: CCAGGGTCTTGAAAGGATCA	239 bp	[35]
TLR4	F3: ATCTTTCAAGGTGCCACATCC R3: ACTGACCTACCGATTGGACAC	194 bp	
MyD88	F4: TTTACAGCATGAATCCCTTGGC R4: TGGGAGTGTAAAATCCTGGTGT	184 bp	
IFNα	F5: TCAGCACCACATCCACCACCTT R5: GGTTCTGGAGGAAGTGTTGGAT	131 bp	
IFNβ	F6: TCAGCAGTCCAAGCATCCCT R6: GGAAGTGTTGGATGCTCCTGAAGTA	186 bp	
IL-6	F7: GGTCCAGAACAACCTCAACCTCC R7: CGTTGCCAGATGCTTTGTGC	202 bp	
IL-8	F8: CGGCATCGGTGTTCTTATCT R8: CTGTCCAGTGCCTTCAGTTT	147 bp	
MDA5	F9: GCTACAGAAGATAGAAGTGTCA R9: CAGGATCAGATCTGGTTCAG	120 bp	[36]
IPS-1	F10: TGCGACCGCCTACAAATTCTAT	139 bp	
	R10: AGGGGTTTGGTAGAGGTCGTAG		
NF-κB	F11: ATCAACCCTTTTAACGTGCCT	142 bp	
	R11: GGTTGGAAATCAAAGGAGGC		
TRIF	F12: TCTACTCACTGCTGGCAAAGG	129 bp	
	R12: CAGCCAGGACGCAGTTTTGTG		
IRF7	F13: ACAACGCCAGGAAGGATGT	120 bp	
	R13: AGCGAAAGTTGGTCTTCCACT		
RIG-I	F14: GCGGATAGAGGCAACAAT	133 bp	
	R14: AGTTATGCCTGCTGCTTT		
DTMUV envelope	F15: AATGGCTGTGGCTTGTTTGG	207 bp	[37]
	R15: GGGCGTTATCACGAATCTA		
U6	F16: CTCGCTTCGGCACGACA	73 bp	[38]
	R16: GCGTGTCATCCTTGCGC		
β-action	F17: CCCCATTGAACACGGTATTGTC R17: GGCTACATACATGGCTGGGG	199 bp	[35]

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
