# Peer review of "DEF Cell-Derived Exosomal miR-148a-5p Promotes DTMUV Replication by Negative Regulating TLR3 Expression"

_viruses, 2020, doi:10.3390/v12010094_

Round 1
Reviewer 1 Report
This paper examined the relative role of TLR3 and MDA5 in immune responses against Duck Tembusu virus and the potential role of exosomal miR-148a-5p in regulating these responses.
Overall, I found this paper somewhat difficult to follow. The justification for various experiments was not fully provided. For example why use DEF? Similar studies examining role of TLR3 and MDA-5 have been performed in chickens (ref 22). However, it is unclear why you would expect significant differences in ducks. Further, after reading the paper, I still do not understand if DEF express TLR3 or MDA-5. Based on Figure 5 it seems like you have to introduce TLR3 and MDA-5 expression (not overexpress) to illicit a response.
The exosome data is an interesting new angle. However, I am not sure how to interpret the data. What is the relevance of DEFs in this context? Is miR-148a-5p expressed/release in exosome by other cells? Also, the authors show that miR-148a-5p is decreased following DTMUV infection but why? It seems somewhat counter intuitive. If the virus was trying to escape TLR3 induced IFN responses, why would it down regulate expression in the cell and the exosome? I am also not sure why the authors chose to “over-express” TLR3 expression on cells to show that TLR3 negatively regulates miR-148a-5p expression. Can the DEF be activated to induce TLR3 expression naturally? What is the application of these findings? How does this fit into the PBMC story?
Other comments
Manuscript needs to be reviewed as there are a number of spaces missing between words
Introduction:
Need further description of ATMUV pathogenesis, which cells are infected etc. Why duck embryo fibroblasts? Need further discussion of reference 22 in introduction Introduction spends too much time explaining the results from the current study
Method:
Improve language in materials and methods section on ethic approval Was approval not required for animal studies? Please provide this information Provide more information on RT-PCR methodology For methods often said as previously described but did not include the reference (example line 210)
Results:
Why are you looking at responses in thymus, spleen, liver? Further justification is required Authors only looked at mRNA expression. What is happening at the protein level? Authors should be assessing cytokine production in supernatant and TLR3/MDA-5 expression via flow What levels of TLR3 and MDA-5 do these cells express at baseline What was your transfection efficiency for TLR3 and MDA5? Please show this data. In control cells (the absence of overexpression of TLR3 or MDA-5), does infection induce an antiviral profile? What happens if you use shRNA in cells in these control cells? How do your results differ from those in reference 22? Knockdown is not complete but you completely lose cytokine mRNA induction? Can you explain this? For IRF3/7 and NF-kb is not enough to look at mRNA – you need to examine protein levels and more specifically phosphorylated protein levels for IRF3, IRF7 and subunits of NF-kB Figure 1 there is no TLR4 data shown (panel H) IFN-alpha and beta are not traditionally called inflammatory cytokines Line 320…..mentions previous study but does not provide the reference
Author Response
The point-by-point response to the reviewer's comments are in the attachement, please check.

Reviewer 2 Report
In this manuscript entitled "DEF cell-derived exosomal miR-148a-5p impairs DTMUV replication by negative regulating TLR3 expression", the authors study the interplay between miRNA 148a-5p and TLR3, and the impact on the replication of duck Tembusu virus, an important flaviviral avian pathogen.
The authors use a variety of techniques, including the isolation of duck exosomes. Although these techniques and the reported results are individually relatively convincing, the interpretation made by the authors is disputable and the scientific reasoning justifying the different step of the study is not so clear.
For instance, in the Abstract section, the authors indicate (lines 30-31) that "duck Tembusu virus can repress miR-148a-5p expression", while (lines 31-32) "miR-148a-5p overpexression impairs duck Tembusu virus replication by negatively regulating ... TLR3 and IFN-b", but they conclude that (lines 34-35) "the negative effect of TLR3 on miR-148a-5p expression may be a strategy for ducks to resist DTMUV infection".
This does not make any sense. Does a reduction in miR-148a-5p expression favor virus replication or duck resistance? It cannot be both.
In addition, given the high virulence of the virus in ducks, this "strategy" used by ducks to resist the infection seems quite inefficient. Whether or not differences exist at this level between duck species or breeds that might account for various levels of resistance to DTMUV was not discussed by the authors.
Several sentences are difficult to understand and would need rephrasing. For instance (lines 118-119), "And the AMTUV we use in this article is biosafety.". Meaning?
Some figures are not consistent with their title. As an example, Figure 1 is entitled "ATMUV-infected ducks effectively upregulate TLR3, MDA5, and inflammatory cytokine expression in the thymus.", but the expression of TLR3 is not shown in Figure 1. In addition, the legend of Figure 1 refers to the mRNA expression of TLR4 (H), while Figure 1H is not represented.
Many figures are very small and difficult to read.
Figure 9, interpreted by the authors as a proof of exosome uptake by PBMCs, is not so clear to me.
Finally, the discussion is much too short. I appreciated the fact that the authors identified limitations to their work, but there are additional limitations. In addition, many points would deserve discussion, especially the implications in terms of host-pathogen interactions. Are there variations in the duck population that could be used to increase the innate resistance to TMUV? Alternatively, would it be possible to use these observations for therapeutic purposes? Are these conclusions applicable only to TMUV or might they apply to related Flaviviruses, including those affecting non avian species?
Author Response

(The authors gave the same response as above.)

Round 2
Reviewer 1 Report
Authors have made a number of the suggested changes which has improved the paper. There are a couple of small typos in the paper. The manuscript can use a thorough reading.
Author Response
|
|
|
Manuscript ID: viruses-626170
Type of manuscript: Original Article
Title: DEF cell-derived exosomal miR-148a-5p promotes DTMUV replication by negative regulating TLR3 expression
Correspondence Authors: Renyong Jia, Anchun Cheng
October 10, 2019
Dear editor :
Thank you very much for your attention and the referee’s evaluation and comments on our paper DEF cell-derived exosomal miR-148a-5p promotes DTMUV replication by negative regulating TLR3 expression. We have revised the manuscript according to your kind advices and referee’s detailed suggestions. Enclosed please find the responses to the referees.
We sincerely hope this manuscript will be finally acceptable to be published on Viruses. Thank you very much for all your help and looking forward to hearing from you soon.
Best regards
Sincerely yours
Renyong Jia
PhD, Professor
Institute for Preventive Veterinary Medicine
College of Veterinary Medicine, Sichuan Agricultural University
Mobile: +86-139-8085-9871
Email: jiary@sicau.edu.cn
Please find the following Response to the comments of referees:
Response to the referee’s comments (all the line means that is in the manuscript with revision marks)
Response to Reviewer 1
Point 1: Authors have made a number of the suggested changes which has improved the paper. There are a couple of small typos in the paper. The manuscript can use a thorough reading.
Response 1: Thans for the reviewer’s kind suggestion. We carefully read and revised the article again, corrected the spelling, grammar and other mistakes in the article.

Reviewer 2 Report
I thank the authors for their response.
However, I have serious concerns about the study.
In spite of my specific demands, there are still contradictions in the manuscript. As an example, the authors state that "miR-148a-5p overexpression impairs DTMUV replication by negatively regulating the
expression of TLR3", meaning that a downregulation of TLR3 impairs viral replication, while they conclude that TLR3 overexpression might confer a protection against viral infection. TLR3 either inhibits or favors viral replication. This clarification was already asked in the first review report.
Besides, what is the biological (in terms of host-pathogen interactions) relevance of the study? The authors argue that TLR3 might be critical in the resistance of ducks to DTMUV, but the fact is that ducks are highly sensitive to the infection. There is thus no question about a so-called resistance. Again, this point had already been highlighted in the first review.
The English language requires extensive editing and in my opinion a proper reviewing is not possible in the current state of the manuscript. Similarly, the response report provided by the authors is poorly written (in terms of English language), and I think the authors sometimes just did not get the meaning of my comments.
I am firmly convinced that there is a lot of laboratory work behind this study. But the scientific soundness of the study design (justification of the methods used and how the different parts of the study are linked to each other), of the introduction and of the discussion, are currently insufficient. I believe that a proper language editing would probably considerably improve that.
Lastly, I would recommend the authors to adapt their conclusions to the results and according to the current knowledge about the host-pathogen interactions of DTMUV in ducks.
Author Response
Manuscript ID: viruses-626170
Type of manuscript: Original Article
Title: DEF cell-derived exosomal miR-148a-5p promotes DTMUV replication by negative regulating TLR3 expression
Correspondence Authors: Renyong Jia, Anchun Cheng
October 10, 2019
Dear editor :
Thank you very much for your attention and the referee’s evaluation and comments on our paper DEF cell-derived exosomal miR-148a-5p promotes DTMUV replication by negative regulating TLR3 expression. We have revised the manuscript according to your kind advices and referee’s detailed suggestions. Enclosed please find the responses to the referees.
We sincerely hope this manuscript will be finally acceptable to be published on Viruses. Thank you very much for all your help and looking forward to hearing from you soon.
Best regards
Sincerely yours
Renyong Jia
PhD, Professor
Institute for Preventive Veterinary Medicine
College of Veterinary Medicine, Sichuan Agricultural University
Mobile: +86-139-8085-9871
Email: jiary@sicau.edu.cn
Please find the following Response to the comments of referees:
Response to the referee’s comments (all the line means that is in the manuscript with revision marks)
Response to Reviewer 2
Point 1: In spite of my specific demands, there are still contradictions in the manuscript. As an example, the authors state that "miR-148a-5p overexpression impairs DTMUV replication by negatively regulating the expression of TLR3", meaning that a downregulation of TLR3 impairs viral replication, while they conclude that TLR3 overexpression might confer a protection against viral infection. TLR3 either inhibits or favors viral replication. This clarification was already asked in the first review report.
Response 1:First of all, thank you for your advice. You have pointed out many times that our conclusion is contrary to the experimental result, but in our opinion, our experimental results are consist to the conclusion, which can be seen from the following aspects.
TLR3, an immune factor in ducks, does not inhibit the virus or promote its replication. However, activation of TLR3 can promote the production of large amounts of IFN-b, which we can see from Figure 9C. Moreover, we found the high level of TLR3 can also inhibit the viral replication process as we can see from Figure 10C. Then our results show that IFN-b can inhibite the viral replication process in a time - and dose-dependent manner, as we can see from Figure 11. In our view, this is the reason why high level of TLR3 can inhibite the viral replication. Therefore, our results suggest that increased TLR3 expression and activation of TLR3 can promote the expression of IFN-b, which negative affecting the viral replication process. And in our article, we made two conclusions, one is MDA5 and TLR3-dependent signaling pathways have important roles in immunity during DTMUV infection (line 446), the other is Type I IFN significantly impairs DTMUV replication in dose- and time-dependent manners (line 503). And when we summarize these two results, we conclude that TLR3 overexpression can negative affect DTMUV replication process via induce abundant of IFN-b.
In our study, miR-148a-5p can target TLR3 and negative regulate the expression of TLR3, which means it can promote DTMUV replication by negatively regulating the expression of TLR3. And this information we can see in the Title and the conclusion part.
Point 2: Besides, what is the biological (in terms of host-pathogen interactions) relevance of the study? The authors argue that TLR3 might be critical in the resistance of ducks to DTMUV, but the fact is that ducks are highly sensitive to the infection. There is thus no question about a so-called resistance. Again, this point had already been highlighted in the first review.
Response 2: Thanks to the reviewer for this pertinent opinion. We acknowledge that ducks are susceptible to DTMUV so far, but this does not contradict our experimental results or conclusions.
Firstly, as we mentioned earlier, TLR3 has important immunological effects, mainly manifested in its ability to produce large amounts of IFN-b that negatively affect the replication process of DTMUV. In other words, TLR3 has function of resisting DTMUV infection. But all of these results come from laboratory experiments, the ducks may not be able to produce the same amount of TLR3 that is artificially overexpressed. For example, in our study, we overexpressed the TLR3, but it might not possible for ducks to produce such large amount of TLR3. Thus, its function to resist DTMUV might not be as significant in ducks, that’s might the reason why even DTMUV can trigger robust native immunity in ducks, DTMUV can still successful invade ducks.
Secondly, even if the ducks were able to generate large amounts of IFN-b by stimulating TLR3 to resist the invasion of the virus, this does not mean that the high expression of TLR3 can prevents ducks infect DTMUV, because they're two separate processes. Whether ducks are highly sensitive to the infected is related to the virus, however the duck's ability to resist the virus invasion is related to the body's immune system. And our experiments only reveal the mechanism by how the duck's immune system fights off the virus.
To the sum, we believe our conclusion that TLR3 is critical in the resistance of ducks to DTMUV is not contract to the fact that ducks are highly sensitive to the infection.
Point 3: I am firmly convinced that there is a lot of laboratory work behind this study. But the scientific soundness of the study design (justification of the methods used and how the different parts of the study are linked to each other), of the introduction and of the discussion, are currently insufficient. I believe that a proper language editing would probably considerably improve that.
Response 3: Thanks to the reviewer for his/her affirmation of our experimental work. Although we illuminate experimental ideas in the introduction and discussion (line 86-108, line 115-131, line 663-668, line 680-710 ), we are happy to introduce our experimental ideas to the reviewer again.
Our experiment is mainly consists of three parts, and there are close connections. In the first part we explored which immune factors are important in the antiviral process. In the second part, we explore the immune function of these immune factors. In the third part, we explore how the expression of these immune factors is regulated by exo-miRNA. It provides a basis for deeper study of the mechanism of anti-virus.
Moreover, we referred to a lot of literature to design our experiment,and these literatures appear in our paper as the theoretical basis of our experiment. First of all, we referred to the literature of S. Chen.; G. Luo.; Y. Zhou.; et al. Avian Tembusu virus infection effectively triggers host innate immune response through MDA5 and TLR3-dependent signaling pathways. Vet. Res 2016, 47, 74, Y.T. Tsai.; S.Y. Chang.; et al. Human TLR3 recognizes dengue virus and modulates viral replication in vitro. Cell. Microbiol 2009, 11, 604–615, N. A. Dalrymple.; V. Cimica.; E. R. Mackow. Dengue virus NS proteins inhibit RIG-I/MAVS signaling by blocking TBK1/IRF3 phosphorylation: dengue virus serotype 1 NS4A Is a unique interferon-regulating virulence determinant. Mbio 2015, 6, 00553–00515 and so on to carry out our experiment in part 1 and part 2. S. Pinto.; C. Cunha.; M. Barbosa.; et al. Exosomes from NSC-34 cells transfected with hSOD1-G93A are enriched in miR-124 and drive alterations in microglia phenotype. Front. Neurosci 2017, 11, 273, B. Zhang.; A. Wang.; C. Xia.; et al. A single nucleotide polymorphism in primary-microRNA-146a reduces the expression of mature microRNA-146a in patients with Alzheimer's disease and is associated with the pathogenesis of Alzheimer's disease. Mol. Med. Rep 2015,12, 4037–4042, M. Zhou.; J. Chen.; L. Zhou.; et al. Pancreatic cancer derived exosomes regulate the expression of TLR4 in dendritic cells via miR-203. Cell. Immunol 2014, 292, 65 –69 and so on are used for reference to design the exploration of biological functions of exosomes.
Moreover, in order to enrich our article, we emphasize our experimental ideas, experimental basis and other information again in the discussion part.
Point 4: Lastly, I would recommend the authors to adapt their conclusions to the results and according to the current knowledge about the host-pathogen interactions of DTMUV in ducks.
Response 4: Thanks for the viewer’s kind suggestion. We researched and learned the information about the host-impact interactions of DTMUV in ducks. We then integrated this information into our paper, which reviewers can see from article line 731-760.
